# Absence of quantum features in sideband asymmetry

J.D.P. Machado[1*], Ya.M. Blanter[1]

**1** Kavli Institute of Nanoscience, Delft University of Technology, Lorentzweg 1, 2628 CJ Delft, The Netherlands
*m2501@gmx.com

January 6, 2020

## Abstract

Sideband asymmetry in cavity optomechanics has been explained by particle creation and annihilation processes, which bestow an amplitude proportional to 'n+1' and 'n' excitations to each of the respective sidebands. We discuss the issues with this interpretation and why a proper quantum description of the measurement should not display such imbalance. Considering the case of linearly coupled resonators, we find that the asymmetry arises from the backaction caused by the probe and the cooling lasers.

## 1  Introduction

Biased misconceptions often become dogmas provided that a blurry experimental connexion is found. It is thus with the quantum interpretation of sideband asymmetry.

Sideband asymmetry (SA) refers to the difference in the spectral height of the side peaks accompanying a drive frequency. When a system is driven coherently at a frequency $\omega_L$ and it is coupled to an oscillator (such as a mechanical resonator), the spectrum acquires peaks (the sidebands) at $\omega_L \pm \Omega$, with $\Omega$ the mechanical frequency. This phenomenon was

first observed with trapped ions [1, 2] and neutral atoms [3], where laser cooling unveiled motional sidebands around atomic transitions. With the emergence of optomechanics, SA was observed in systems with larger mechanical elements such as nanobeams [4, 5], LC-resonators [6], ultracold atoms [7], and membranes [8, 9]. In the absence of any symmetry breaking mechanism, it would be expectable for the sidebands to be equal. However, experimental observations reveal that one sideband is larger than the other. This imbalance was justified by an asymmetric role of zero-point motion (ZPM) in the computed spectrum [10–14]. Such quantum exegesis originates from proclaiming *ex cathedra* that the measurement outcome is described by

$$S_{XX}(\omega) = \int_{\mathbb{R}} e^{i\omega t} \langle \hat{x}(t)\hat{x}(0)\rangle_{th} dt = \delta(\omega + \Omega)\bar{n}_{th} + \delta(\omega - \Omega)(\bar{n}_{th} + 1) \,, \qquad (1)$$

where $x$ is the displacement of the oscillator, and $\bar{n}_{th}$ its thermal occupancy. By identifying $\pm\Omega$ with the sidebands, SA would be naturally explained by ZPM, and consequently prove the quantum nature of the mechanical element (regardless of its state). Thus, by cooling the resonator sufficiently, the asymmetry as well as the quantum nature of the macroscopic resonator would become visible, and no classical theory could explain this phenomenon [4]. Furthermore, SA promised an experimental paradise where temperature could be determined without any calibration [7–9]. Following such experimental observations, the quantum nature of SA was deemed true.

The desire to observe quantum effects at the macroscopic scale was stimulated by developments in nanomechanics, and it raised the question of where does the quantum realm frontier lie. This question has increased the necessity for a strict limit, where past a given borderline, certain phenomena would necessarily have a fundamental quantum nature. Though it remains unanswered, it led to the recognition that the use of an operator formalism does not imbue a quantum nature for the system under consideration, and it allowed for deeper analyses on the nature of SA. Alternative interpretations and explanations such as interference between different noise channels [6, 15] and laser phase noise [16] have been discovered. Despite certain flaws of the new interpretations [1], the claim of a quantum nature for SA became disputable [17].

Regarding SA, a pervasive problem plagues its interpretation: *a priori* definitions. The interpretation of SA differs for different operator orders (arising from different detector models [18]), and to assert that by definition, the experimental apparatus measures a particular operator order, does not force a detector to measure that specific order. Theoretical interpretations should be based on the physical situation, instead of the experimental validation being subdued to theoretical postulates. This constitutes a problem for experimental validation, as biased premises have been the starting point.

In this article, we start by discussing the problems with the standard interpretation of SA, their connection to how measurements are performed, and the role of ZPM in the spectrum (Sec. 2). We proceed to compute the response function for a system composed by two driven optical modes and a mechanical one (Sec. 3). Considering the symmetric noise power spectral density for both cases, we show that ZPM does not contribute to the asymmetry and that SA naturally arises from the backaction between the cavity and the mechanical oscillator. Additional details of the calculations can be found in App. A. For comparison with other measurement methods, we also analyse the case of a system composed by two modes (cavity + mechanical), but driven with multiple tones in App. B. Finally, we present a concise review of previous analyses of SA in App. C.

---

[1] see App. C for details

## 2    The measurement problem

The problem over the nature of SA can be traced back to its measurement. In contrast to its classical counterpart, defining the power spectral density in a quantum framework poses a problem regarding the operators' order. Direct substitution of the fields by operators in classical formulas is dangerously arbitrary, as there is a multitude of possibilities and not all of them have a physical meaning. The problems with defining a quantum spectral density and the meaning of the distinct possibilities were raised before [17, 19] but remain unsolved. The usual way to address these issues is to take the specific measurement procedure into consideration. SA can be measured using well-known linear detection schemes such as homodyne or heterodyne detection. The quantum description of these techniques [20–22] typically focuses on the quadrature measurement and noise response on the time-domain, leaving issues with the frequency domain unmentioned. To measure the field quadrature $X(t) = a_s(t) + a_s^{\dagger}(t)$ (where $a_s$ is the annihilation operator for the signal), linear detection schemes combine the signal input with a local oscillator, and split them into two detectors. The intensity difference between the detectors is proportional to $X(t)$, but it is in the frequency domain that the noise response is computed via the quadrature variance. This poses the problem of defining the variance of $X(\omega)$. As $X(\omega)$ is a complex operator, there are different possible orderings, such as $(\#1)\langle (X(\omega))^{\dagger} X(\omega) \rangle$ or $(\#2)\langle X(\omega)(X(\omega))^{\dagger} \rangle$ as well as any convex linear combination. Each possibility produces a different outcome, but the uniqueness of the spectrum implies that only one should represent the observed spectrum. The noise power spectral density is obtained with the Fourier transform of the average of the product between different measurement outcomes. As $X(t)$ is hermitian, the measurement outcome is a real number, and so is the product at different times. However $X(t)X(t')$ is not strictly hermitian, and therefore it can have non-real values as a possible outcome. Therefore $\langle X(t)X(0) \rangle$ cannot represent the physical measurement, and so can neither Eq.(1). The only hermitian possibility that can represent the measurement is the symmetric combination of $X(t)X(t')$ with its hermitian conjugate. Therefore, the most suitable spectral density to describe the measurement is

$$\bar{S}_{XX}(\omega) = \frac{1}{2}\Big\langle X(\omega)X(-\omega) + X(-\omega)X(\omega) \Big\rangle. \tag{2}$$

An alternative way to measure SA is with photodetection, and for this case the ordering issues are usually bypassed by choosing a detector model and establishing a link with the measurement outcomes. The typical detector model consists of a single qubit interacting briefly with the measured field via a weak dipolar coupling [23, 24]. The excitation probability $P_{exc}$ of a qubit in the ground state for short time-scales and coupled to a stationary random field can be computed with perturbation theory, and it is [19]

$$P_{exc} \propto \int_{-t}^{t} e^{i\epsilon t'} \langle X(t')X(0) \rangle dt'. \tag{3}$$

By identifying the qubit energy splitting $\epsilon$ with the frequency $\omega$, and $P_{exc}$ with the measured signal, Eq.(3) has been employed as a quantum spectral density. However, such toy model is unable to completely model the measurement because: (1) spectrometers are not composed of a single qubit, and a single qubit alone cannot provide the spectral density for a wide frequency range. Models with several qubits lead to higher order correlation functions [24] and higher spin states do not lead to Eq.(3) [25]; (2) Eq.(3) is valid for short time-scales, where the transition rate is a constant given by the Fermi golden rule. To obtain the spectral density, the system has to be monitored for extended time-intervals, after which the validity of this result breaks down; (3) other detection models lead to different operator orders, such as anti-normal order in photon counters [26].

Irrespectively of the model and definitions considered, ZPM should not play a physical role in the asymmetry. Even though the measured field quadrature is associated with the operator $X$, the outcome of a measurement is a scalar $x$, and it is with the measurement record $x(t)$ that the spectrum is obtained. For the scalar $x(t)$, the order issue does not exist, the spectral density is well-defined, and there is no reason for ZPM to affect the sidebands differently. Nevertheless, ZPM plays a role in the variance of $X$, and there is a link between $X(t)$ and the measurement outcome. As $X$ is monitored in time, a definite proof might rest in the theory of quantum continuous measurements. The formalism of continuous position measurements already exists [27,28], as well as analogous formalisms to model photodetection [29]. However, we are unaware of similar approaches to describe the spectral density. A closely related approach to describe homo- and heterodyne detection featuring quantum trajectories is also available in the literature [30] but such approach still relies on operator order postulates to evaluate the spectrum and not solely on the measurement record.

## 3  Multimode measurement

For the reasons exposed above, Eq.(2) shall be used to compute the spectrum. To examine the nature of SA, we consider the optomechanical case where the sidebands are measured via a signal coming from an optical (or microwave) cavity coupled to a mechanical resonator. From input-output relations, the signal amplitude is proportional to the cavity field, and for linear couplings, the cavity field yields a linear relation with the mechanical displacement. For this reason, when the cavity is driven, the coupling to the mechanical resonator produces sidebands around the drive frequency that contain information about the mechanical motion.

A method to measure SA is to send a probe beam at $\omega_{cav} - \Omega$ and measure the red-sideband at $\omega_{cav}$, and then change the probe frequency to $\omega_{cav} + \Omega$ to measure the blue-sideband at $\omega_{cav}$ (see Fig.1). This way, each sideband is enhanced separately while the other sideband is off-resonant, and SA is measured more easily. Additionally, in a typical experimental situation, the mechanical resonator is cooled via an independent source. This source can be a distinct cavity mode driven at the red-sideband, and this approach shall be analysed ahead. An alternative way to measure SA with only one cavity mode is to simultaneously drive the system with two probe tones (see Fig.1), with these two tones slightly detuned by $\delta$ from $\omega = \omega_{cav} \pm \Omega$ such that the sidebands do not overlap at $\omega_{cav}$. This approach renders quite similar results, and so it will be analysed in App. B.

To better compare with the experimental situation, we consider two cavity modes: a cooling mode, and a read-out mode with a frequency far away from the cooling mode. The equations of motion describing this system are [6]

$$id_t b = \left( \Omega - i\frac{\Gamma}{2} \right) b - \sum_j g_j (a_j + a_j^\dagger) + \eta_b \,, \tag{4}$$

$$id_t a_j = \left( -\Delta_j - i\frac{\kappa_j}{2} \right) a_j - g_j (b + b^\dagger) + \eta_j \,, \tag{5}$$

where $\Gamma$ is the mechanical dissipation and $\Delta_j$, $\kappa_j$, and $g_j$ are the detuning, cavity linewidth and coupling strength for mode $j$. The detuning $\Delta_j = \omega_{L,j} - \omega_{cav,j}$ accounts for the shift of each mode $j$ from their respective drive reference frame, i.e. a reference frame with frequencies displaced from the drive frequencies $\omega_{L,j}$. Here and onwards, the cavity frequency shift produced by the static displacement of the resonator is included in $\omega_{cav,j}$. Furthermore, $b$ represents the phonon annihilation operator and $a_r, a_c$ the photon anni-

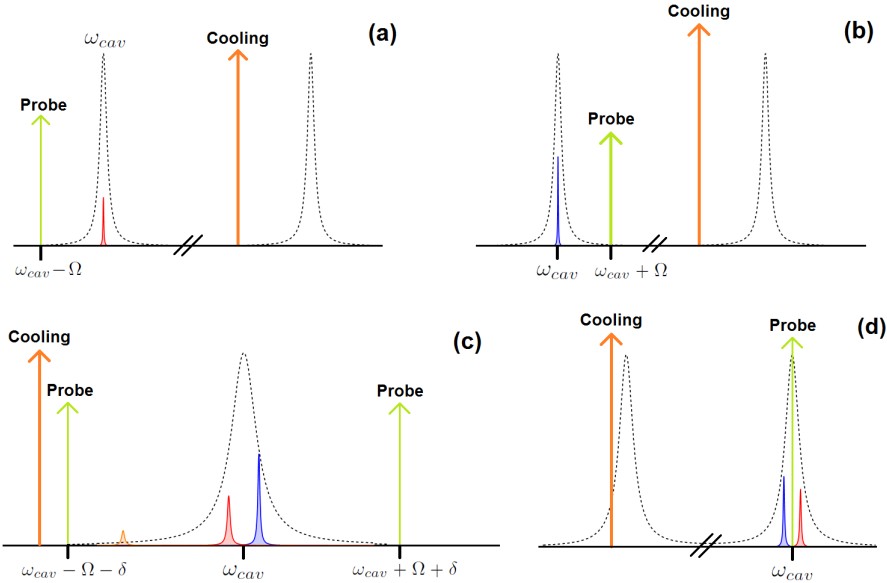

Figure 1: Different schemes to measure sideband asymmetry. The sidebands can be measured one at a time by placing the probe red(blue)-detuned (panel a (b)). Alternatively, a single cavity mode can be probed with 2 tones, which create sidebands within the cavity linewidth (panel c). The sidebands can also be measured directly with a probe tone on resonance (panel d). Cooling tones are also represented for completeness.

hilation operators for the read-out and cooling modes. At last, $\{\eta_j\}$ are the noise terms, with the properties

$$\langle\eta_j^\dagger(t)\eta_l(t')\rangle = \frac{\kappa_j}{2\pi}\bar{n}_j\delta_{jl}\delta(t-t'),\tag{6}$$

$$\langle\eta_j(t)\eta_l^\dagger(t')\rangle = \frac{\kappa_j}{2\pi}(\bar{n}_j+1)\delta_{jl}\delta(t-t'),\tag{7}$$

where $\bar{n}_j$ is the thermal occupancy for mode $j$. An analogous relation holds for the mechanical noise. Note that the system behaves linearly as long as the interaction is weak enough to prevent entering the amplification regime. When this regime is reached, an instability takes place (primarily at $\Delta = \Omega$), leading to a behaviour very different than just the creation of sidebands. Moreover, in the strong coupling regime, hybridisation between the cavity and the mechanics occurs, leading to additional spectral features, such as a frequency splitting at $\Delta = -\Omega$. As we are only concerned in addressing the SA issue, only the weak-coupling regime $g_j < \kappa_j, \Omega$ shall be considered, and since cooling occurs at the red-sideband, we set $\Delta_c = -\Omega$. Performing a Fourier transform in Eqs.(4-5) leads to the linear response function of the systems. The read-out field has the form

$$a_r(\omega) = q_1\eta_r(\omega) + q_2[\eta_r(-\omega)]^\dagger + q_3\eta_B(\omega) + q_4[\eta_B(-\omega)]^\dagger + q_5\eta_c(\omega) + q_6[\eta_c(-\omega)]^\dagger,\tag{8}$$

where the coefficients $q_j$ are displayed in App. A, along with the cavity response for the case where each sideband is driven separately.

In general, the read-out field does not have the same intensity at the red- and blue-sidebands because of the backaction from the cooling and read-out modes. This can be verified by evaluating, for example, the case with $\Delta_r = 0$ (corresponding to the experimental situation in [8,9]) in the limit $\Gamma \ll g_j, \kappa_j, \Omega$, which gives

$$\left|\frac{q_1(\Omega)}{q_1(-\Omega)}\right| = \left|\frac{A_- + iB_-}{A_+ - iB_+}\right| \neq 1,\tag{9}$$

where $A_\pm \approx 2(1+C_c)\Omega - \frac{\kappa_2}{4\Omega}(\kappa_r \pm \kappa_c C_r)$, $B_\pm \approx \kappa_r(1+C_c) + \kappa_c\left(\frac{1}{2} \pm C_r\right)$, and $C_j = \frac{4g_j^2}{\Gamma\kappa_j}$ is the cooperativity for mode $j$. Thus, the asymmetry does not present a method for absolute self-calibrated thermometry.

Using Eqs. (2) and (19-23), the spectral density for each enhanced sideband is found to be

$$\bar{S}_{XX}^\pm \approx \frac{(1+C_c)^2 + \frac{\kappa_r^2}{16\Omega^2}C_r^2}{(\omega \pm \Omega)^2 + \frac{\kappa_r^2}{4}(1+C_{eff}^\pm)^2}\kappa_r(\bar{n}_r + 1/2) + \frac{4\xi_r^2\Gamma(\bar{n}_B + 1/2 + C_c(\bar{n}_c + 1/2))}{(\omega \pm \Omega)^2 + \frac{\Gamma^2}{4}(1+C_{eff}^\pm)^2}, \quad (10)$$

where $X = a + a^\dagger$ and $\pm$ correspond to the blue(+) or red(−) sidebands. As seen from Eq.(10), the only difference in the expression for the sidebands lies in the denominator, where the interaction gives a different contribution for the linewidth of each sideband. It is also clear that ZPM does not contribute to the imbalance, and that the origin of the asymmetry for the weak-coupling and resolved sideband regime is the distinct effective optomechanical dampings for each sideband. Thus, instead of the height of the sidebands being given by $n$ and $n + 1$, we find that the height of both sidebands is proportional to $n + 1/2$, with $n$ the number of thermal excitations. A consequence of this difference is that at $T = 0$, the red sideband does not vanish. One could think that at $T = 0$, the absence of phonons would prevent the sideband to exist. However, as we are considering quadrature measurements instead of photon counting, the ZPM of the resonator affects the measurement of its position, and allows for the existence of the red sideband.

The asymmetry is quantified experimentally via the noise power $I^\pm$, which is obtained by integrating the area of the resonant sidebands $S^\pm$ over all frequencies. The asymmetry factor $\zeta$ quantifying the imbalance is given by

$$\zeta = \frac{I^+}{I^-} - 1 = \frac{2C_r}{1 - C_r + C_c} = \frac{8g_r^2\kappa_c Q}{\kappa_r\kappa_c\Omega - 4(g_r^2\kappa_c - g_c^2\kappa_r)Q} . \quad (11)$$

As backaction already occurs at the classical level, Eq.(11) shows a classical origin for SA. For real physical systems, the mechanical quality factor $Q$ varies with the temperature, which leads to a temperature dependent asymmetry. For small cooperativities, the asymmetry is directly proportional to the quality factor. Whenever the quality factor decreases linearly with increasing temperature (at least for a given temperature range), then the qualitative temperature dependence of the asymmetry matches the standard result.

# 4   Conclusion

In conclusion, the undisputed existence of SA is not a proof of any quantum nature of the system. By computing the symmetric noise power spectral density for a system of linearly coupled oscillators, we have shown that SA arises from the backaction caused by the laser drive, and that no ZPM contributes to the asymmetry. The symmetric spectral density has already been employed in [6,17], but the asymmetry was attributed to interference between the cavity noise and the mechanical resonator's noise. Such misinterpretation sprouts from miscalculations (see Appendices B and C). Here, only white noise was considered, but the analysis can be extended for any type of noise. Nevertheless, we expect coloured noise to be unimportant since only frequencies within a narrow bandwidth of $\Gamma$ contribute to the peak height.

Although our analysis is restricted to coupled harmonic oscillators, the same procedure can be extended to analyse the trapped ions and neutral atoms case [1–3]. The role of backaction has not yet been investigated in these systems, and a thorough analysis would

clarify the nature of the asymmetry for this case. Note that for the case of trapped ions and atoms, their electronic quantum nature may lead to quantum signatures for the output light which are not an intrinsic feature of the light field (neither of the mechanical motion), much like in the case of Raman scattering [31]. Further, our analysis is focused only on the power spectral density obtained with linear detections methods. There are other methods closely related to SA which also make use of the term "sideband asymmetry", such as the difference between the photon count rates when an optomechanical system is driven with short pulses at the red and blue sidebands [32, 33], that are outside the domain of the present work. Because single photon detection is a method that does not exhibit the operator order issues raised in Sec.2, and because only the instantaneous count rate is measured instead of the frequency spectrum (for weak driving and short enough pulses, the effects of backaction and noise are negligible), our conclusions do not apply to these single photon detection methods. The quantum description of the photon count detection scheme does indeed display a $n/(n+1)$ imbalance and can be used to directly determine the phonon occupancy. Nevertheless, it would be interesting to understand how the two methods are related to each other.

We stress once more that there are no quantum features in SA. For a linear system, the noise response function is identical in both classical and quantum descriptions, and as Stokes and anti-Stokes processes provide different amplitudes for the sidebands, an asymmetry naturally emerges.

## Acknowledgements

We thank G. Welker, C. Schäfermeier, and S. Sharma for the useful feedback provided.

**Funding information** This work was supported by the Dutch Science Foundation (NWO/FOM).

## A Fourier coefficients and optical response function

The Fourier coefficients $\{q_j\}$ displayed at Eq.(8) are given by

$$q_1 = \beth(\omega)\left[\left(\Omega^2 - (\omega + i\frac{\kappa_c}{2})^2\right)\left(\left(\Omega^2 - (\omega + i\frac{\Gamma}{2})^2\right)(\Delta_r - \omega - i\frac{\kappa_r}{2}) + 2\Omega g_r^2\right)\right.$$
$$\left. - 4g_c^2\Omega^2\left(\Delta_r - \omega - i\frac{\kappa_r}{2}\right)\right], \tag{12}$$

$$q_2 = -\beth(\omega)2\Omega g_r^2\left(\Omega^2 - \left(\omega + i\frac{\kappa_c}{2}\right)^2\right), \tag{13}$$

$$q_3 = \beth(\omega)g_r\left(\Delta_r - \omega - i\frac{\kappa_r}{2}\right)\left(\Omega + \omega + i\frac{\Gamma}{2}\right)\left(\Omega^2 - \left(\omega + i\frac{\kappa_c}{2}\right)^2\right), \tag{14}$$

$$q_4 = \beth(\omega)g_r\left(\Delta_r - \omega - i\frac{\kappa_r}{2}\right)\left(\Omega - \omega - i\frac{\Gamma}{2}\right)\left(\Omega^2 - \left(\omega + i\frac{\kappa_c}{2}\right)^2\right), \tag{15}$$

$$q_5 = \beth(\omega)2\Omega g_r g_c\left(\Delta_r - \omega - i\frac{\kappa_r}{2}\right)\left(\Omega + \omega + i\frac{\kappa_c}{2}\right), \tag{16}$$

$$q_6 = \beth(\omega)2\Omega g_r g_c\left(\Delta_r - \omega - i\frac{\kappa_r}{2}\right)\left(\Omega - \omega - i\frac{\kappa_c}{2}\right), \tag{17}$$

where

$$
(\beth(\omega))^{-1} = \left(\Omega^2 - \left(\omega + i\frac{\Gamma}{2}\right)^2\right)\left(\Delta_r^2 - \left(\omega + i\frac{\kappa_r}{2}\right)^2\right)\left(\Omega^2 - \left(\omega + i\frac{\kappa_c}{2}\right)^2\right)
$$
$$
+ 4\Omega\Delta_r g_r^2\left(\Omega^2 - \left(\omega + i\frac{\kappa_c}{2}\right)^2\right) - 4\Omega^2 g_c^2\left(\Delta_r^2 - \left(\omega + i\frac{\kappa_r}{2}\right)^2\right). \qquad (18)
$$

With these coefficients, we can evaluate the case where each sideband is driven separately. At the enhanced red-sideband+cavity peak, the field amplitude for the read-out mode at $\Delta_r = -\Omega$ is

$$
a_r(\omega) \approx Q_- \eta_r(\omega) - R_-\left(\eta_B(\omega) - 2i\xi_c\eta_c(\omega)\right), \qquad (19)
$$

while at the enhanced blue-sideband+cavity peak, the field amplitude of the probe at $\Delta_r = \Omega$ is

$$
a_r(\omega) \approx Q_+ \eta_r(\omega) - R_+\left([\eta_B(-\omega)]^\dagger - 2i\xi_c[\eta_c(-\omega)]^\dagger\right), \qquad (20)
$$

with $\xi_j = g_j/\kappa_j$ and

$$
Q_\pm \approx \frac{\left(1 - i\frac{\kappa_r}{4\Omega}C_r + C_c\right)}{\omega \pm \Omega + i\frac{\kappa_r}{2}(1 + C_{eff}^\pm)}, \qquad (21)
$$

$$
R_\pm \approx \frac{2i\xi_r}{\omega \pm \Omega + i\frac{\Gamma}{2}(1 + C_{eff}^\pm)}, \qquad (22)
$$

$$
C_{eff}^\pm = C_c \mp C_r, \qquad (23)
$$

and considering the limit $g_j \ll \kappa_j \ll \Omega$ (weak coupling and resolved sideband regime).

## B  Multitone measurement

We analyse here the case where the sidebands are created simultaneously by independent driving tones. For the sidebands to be enhanced, they must fall within the cavity linewidth, but in order to make them distinguishable, the tones must be slightly detuned from the red and blue sidebands by an amount $\delta$ (with $\Gamma \ll \delta \ll \kappa$ so that the peaks do not overlap and remain within the cavity linewidth). To fully describe the experimental situation, we consider an additional cooling tone of frequency $\omega_{cav} - \Omega - \delta_c$, with $\delta_c \gg \delta + \Gamma$ so that the cooling tone does not overlap with the red sideband probe tone. In this multi-tone case, the appearance of beats between the different tones $\varpi_j$ is inevitable, and the linear interaction of Eq.(5) can no longer be made time-independent. Thus, the original form of the interaction must be considered, and the equations of motion for the system are now [6]

$$
id_t a = \left(\omega_{cav} - i\frac{\kappa}{2}\right)a - g_0(b + b^\dagger)a + \eta_A(t) + \sum_j s_j e^{-i\varpi_j t}, \qquad (24)
$$

$$
id_t b = \left(\Omega - i\frac{\Gamma}{2}\right)b - g_0 a^\dagger a + \eta_B(t), \qquad (25)
$$

where $\{s_j\}$ are the tones' amplitudes, and the sum in $j$ is over the $\varpi_j$ frequencies $\{\omega_{cav} \pm (\Omega + \delta), \omega_{cav} - \Omega - \delta_c\}$. The driving terms can be removed with the shift

$$
a(t) = A(t) + \sum_j e^{-i\varpi_j t}\alpha_j \quad , \quad b = \beta + B(t), \qquad (26)
$$

where

$$\alpha_j = \frac{s_j}{\Delta_j + i\frac{\kappa}{2}} \quad , \quad \beta = \frac{g_0}{\Omega - i\frac{\Gamma}{2}} \sum_j |\alpha_j|^2 . \tag{27}$$

With this shift, the resonant part of the interaction is enhanced by $\alpha_j$, and it becomes linear in $A$ and $B$. As $g_0$ is negligible in comparison to the other parameters, the terms linear in $A$ and $B$ suffice to account for the effects of the interaction. Disregarding the nonlinear terms, and performing a Fourier transform at the equations of motion, we find the cavity field to be

$$A(\omega)\mathcal{D}(\omega) = \beth(\omega) - \sum_{p \neq q} \alpha_q^* G(\omega - \varpi_p) A(\omega - \varpi_p + \varpi_q) - \sum_{p,q} \alpha_q G(\omega - \varpi_p)(A(\varpi_p - \omega + \varpi_q))^\dagger , \tag{28}$$

where

$$\beth(\omega) = -\sum_q \left( \frac{g_0 \alpha_q \eta_B(\omega - \varpi_p)}{\omega - (\varpi_p + \Omega) + i\frac{\Gamma}{2}} - \frac{g_0 \alpha_q (\eta_B(\varpi_p - \omega))^\dagger}{\omega - (\varpi_p - \Omega) + i\frac{\Gamma}{2}} \right)$$
$$- \eta_A(\omega) - \sum_{p,r \neq q} g_0 G(\omega - \varpi_p) \alpha_q^* \alpha_r \delta(\omega - \varpi_p + \varpi_q - \varpi_r) , \tag{29}$$

$$\mathcal{D}(\omega) = \omega - \omega_{cav} + i\frac{\kappa}{2} - \sum_q \alpha_q^* G(\omega - \varpi_q) , \tag{30}$$

$$G(\omega - \varpi_p) = \frac{2g_0 \alpha_p \Omega}{\Omega^2 - (\omega - \varpi_p + i\frac{\Gamma}{2})^2} . \tag{31}$$

Eq.(28) provides a general framework for the cavity spectrum when linearly coupled to another oscillator and driven by multiple tones. Obtaining an analytical solution for this system is impractical, because the presence of several tones implies that all arbitrary integer combinations of the tones' frequencies must be evaluated. However, the higher harmonics are off-resonant and they can be disregarded in the weak-coupling regime. Using Eqs.(28-31), the same procedure leads to an equation analogous to Eq.(10), with an effective cooperativity of

$$C_{eff}^\pm = C_c \frac{\Gamma^2}{4(\delta_c \pm \delta)^2 + \Gamma^2} \mp C_\pm \pm C_\mp \frac{\Gamma^2}{16\delta^2 + \Gamma^2} , \tag{32}$$

where $\pm$ stands for the sideband at $\omega_{cav} \pm \delta$. Since $C_{eff}$ is different for each sideband, an imbalance occurs even if the red and blue-sideband tones have the same intensity ($C_+ = C_-$). The difference between $C_{eff}^+$ and $C_{eff}^-$ is due to the $\delta$ frequency shift from the resonance. Thus, identically to the multimode case, backaction leads to an asymmetry in the spectrum.

## C    Concise review

A few analyses of the role of noise in SA have been investigated in the literature in order to ensure that the asymmetry is solely due to ZPM. Here we succinctly review and examine some of them, presented in no particular order. The experiments carried in [6–9] used (homo-)heterodyne detection to measure the sidebands, but the role of backaction was not included. In [15], the system is modeled as two interacting harmonic oscillators, identically to the case considered here. However, the noise response for the mechanical oscillator in the frequency domain (Eq. (5) of [15]) does not follow from the equations of

motion in the frequency domain (Eqs. (3-4) of [15]), and thus it does not account properly for the backaction effects. This affects the 'quantum calibrated asymmetry' presented in Eq. (21) of [15], which is used as a parameter able to distinguish a classical effect from a quantum one, while including backaction [4]. Other problems arise in the analysis of laser phase noise. If the amplitude of an optical laser field ($E_0(t) = |E_0|e^{i\phi(t)}$) undergoes random phase fluctuations ($\phi$ is a random variable with $\langle\phi\rangle = 0$), then $\langle E_0^*(\tau)E_0(0)\rangle \approx |E_0|^2\left(1 - \frac{1}{2}\langle(\phi(0) - \phi(\tau))^2\rangle\right)$, where the approximation holds for small phase fluctuations. This result differs from the one presented between Eqs. (22),(23) of [15], and the difference comes neglecting relevant $2^{nd}$ order terms.

In [4], backaction for the read-out mode is taken into account, but no backaction of the cooling mode is considered. As described in [4], the read-out laser is weaker than the cooling laser, and backaction of the cooling mode should be considered as well, leading to the spectral density in Eq.(10). Moreover, spurious backaction contributions are considered in Eq.(4) of [4] because the mechanical bath occupancy is mistaken with the mechanical phonon number, and improper cooling corrections were applied. The same experiment was also analysed from a theoretical point of view in [10] but considering the case of heterodyne detection. To justify the use of a symmetric spectral density, it is claimed an existent direct connection between heterodyne and photon counting measurements [10] based on the assertion that the amplitude output quadrature $Y$ obeys $[Y(\omega), Y^\dagger(\omega')] = 0$. A direct calculation using Eq.(30) of [10] reveals that this is not correct. The computed spectrum (Eq.(37) of [10]) is obtained using Eq.(36) (of [10]), which is incorrect since the quadratures are not independent. Additionally, backaction from the read-out is explicitly ignored, discarding thus a source of SA.

Despite presenting an alternative interpretation, the explanation of SA as present in [17] is also flawed due to miscalculations. The definition of the spectral density (Eq.(40) of [17]) is always positive, but the results presented (namely Eq.(41)) can be negative. Since the noise sources are uncorrelated, one can derive from the Eq.(40) of [17] that the contribution of each noise source is always positive. A contradiction occurs because the solutions to Eqs.(30-38) do not lead to Eqs.(41,42) of [17].

The same problem is present in [6]. Computing the spectral density as defined in Eq.(2) of [6] using the solutions to the equations of motion does not lead to the spectral densities presented in Eqs.(3-4) of [6]. As a side technical detail, the multitone situation of [6] leads to an explicitly time-dependent linear system (Eqs.(A2a, A2b)), but the analysis is performed assuming that the sideband created by the red-detuned probe can be treated independently from the sideband created by the blue-detuned probe without any proper justification. Further, the computed spectral density (Eq.(A18) of [6]) exhibits corrections to the linewidth of the sidebands, but these are only valid when the tones are precisely on resonance, which can never be the case in order for the sidebands to be resolved. Particularly, if both tones have the same intensity, one is lead to believe that the contributions to the linewidth cancel. However, comparing the total damping presented in [6] with Eq.(32) presented above, one can see that it is not the case because the deviation from resonance is larger than the mechanical linewidth. If both probes are placed on resonance, the linewidths are indeed the ones presented in [6], but the sidebands overlap and it is no longer possible to distinguish the red- from the blue-sideband.

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
