# Peer review of "Absence of quantum features in sideband asymmetry"

_SciPost Physics_

## Round 1 · Referee Report · Anonymous (Referee 1) · 2019-11-4

Strengths

Quite generally, I think this manuscript could be a valuable contribution to the field of quantum optomechanics. Indeed, with regards to experiments showing sideband asymmetry, the authors have convinced me that the question of operator ordering has not been given an appropriate level of attention. I found the paper fairly well written, with relevant examples that are placed in the context of prominent experiments. These comparisons are very pointed and will certainly aid in initiating further discussions on this topic.

Weaknesses

As the paper currently stands, I see two minor weaknesses. Both of which could be easily resolved.

First, the details of the calculations in Sec. 3 detract from the overarching message. For example, the list of Fourier components in Sec. 3.1 (Eq. 9-15) does not, in my opinion, provide any additional insight beyond what is said elsewhere. This is also true for Sec. 3.2 (Eq.28-31), which finds a result similar to that obtained in Sec. 3.1 but with a modified susceptibilty. I suspect a more concise version would actually have a stronger impact (with the details moved into the supplemental material).

Second, there is no reference to recent optomechanical experiments that use single photon detection, such as those performed in the laboratories of Oskar Painter and Simon Groblacher. I understand this is an different measurement technique to homodyne and heterodyne, and not necessarily the target of the claims made in this paper. However, that sub-community has borrowed the term "sideband asymmetry" to denote the difference in click rates between the upper and lower sidebands. Furthermore, they reference many of the papers under discussion in this manuscript. I think it would be valuable to broaden the context and directly comment on these experiments.

Report

As a broad field of research, cavity optomechanics has been around for many decades. However, many early experiments where limited to the classical regime since reaching/measuring the quantum level was (and still is) experimentally challenging. As a result, the community identified certain experimental signatures that could be used to distinguish quantum behaviour from classical behaviour. One such signature is the observation of asymmetry in the displacement noise power spectral density.

In the manuscript presented here, the authors present an argument against this convention, claiming that this particular experimental signature (sideband asymmetry) is an artifact of the decision to use a particular operator ordering. Indeed, I agree with the authors claim that, in contrasting quantum from classical behaviour, one needs to carefully consider the form of measurement being performed (in addition to the physical system itself).

If this paper is published it would hopefully initiate further discussions. Those discussions may, or may not, fall in favour of the claims made here; eitherway I think it's conversation worth having.

I recommend its publication in SciPost

Requested changes

As per earlier suggestions;
1-Condense theoretical section.
2-Add discussion on photon counting experiments.

  • validity: good
  • significance: high
  • originality: high
  • clarity: high
  • formatting: good
  • grammar: good

Author:  João Machado  on 2020-01-09  [id 702]

(in reply to Report 1 on 2019-11-04)
Category:
remark
answer to question

We thank referee 1 for his valuable feedback.

The referee says "First, the details of the calculations in Sec. 3 detract from the overarching message. For example, the list of Fourier components in Sec. 3.1 (Eq. 9-15) does not, in my opinion, provide any additional insight beyond what is said elsewhere. This is also true for Sec. 3.2 (Eq.28-31), which finds a result similar to that obtained in Sec. 3.1 but with a modified susceptibility. I suspect a more concise version would actually have a stronger impact (with the details moved into the supplemental material)."

Our reply: We fully agree with the referee and we have implemented his suggestions. For a description of the new structure of the manuscript, see the list of changes.

The referee says "Second, there is no reference to recent optomechanical experiments that use single photon detection, such as those performed in the laboratories of Oskar Painter and Simon Groblacher. I understand this is an different measurement technique to homodyne and heterodyne, and not necessarily the target of the claims made in this paper. However, that sub-community has borrowed the term "sideband asymmetry" to denote the difference in click rates between the upper and lower sidebands. Furthermore, they reference many of the papers under discussion in this manuscript. I think it would be valuable to broaden the context and directly comment on these experiments."

Our reply: We agree with the referee that the aforementioned single photon detection experiments should be mentioned. However, as the referee is aware, these experiments are not equivalent to the homo-/heterodyne detection considered in our manuscript, and we fear that an extended discussion of single-photon detection is beyond the scope of our manuscript and that it would harm the conciseness and focus we intended to achieve. We have thus briefly discuss them in the manuscript and enlightened the differences. For the additional discussion, see the list of changes.

---

## Round 1 · Referee Report · Anonymous (Referee 2) · 2019-12-22

Weaknesses

1 - Inappropriate language
2 - Unclear on subtle aspects of calculations, while asserting that other work is unclear/incorrect
3 - Unconvincing conclusions

Report

Sideband asymmetry in optomechanical systems has been observed in a number of experiments. The origin of this sideband asymmetry has been discussed by a number of authors, and is somewhat subtle, being dependent on the method of detection of scattered radiation. It has been attributed to zero-point fluctuations of the mechanical or electromagnetic modes, or due to correlation between backaction noise and the imprecision noise of the detector.

The present paper appears to support the latter interpretation, though in a manner which is rather unconvincing and much less clear than prior work on the subject. Here the asymmetry appears to be attributed to “coincidental” variations in intrinsic system parameters, in a manner which is unlikely to be reproducible across multiple experiments and would be experimentally testable, should anybody be sufficiently motivated.

The spectral densities are defined in an unclear manner, presenting a spurious distinction between the time and frequency domain, where in fact they can be related in a well-defined manner if one is sufficiently careful. This has been done in a number of other places.

While I have not attempted to reproduce the authors’ calculations myself, I am skeptical as to their correctness.

Some specific comments follow:
- The authors claim, using melodramatic and inappropriate language, that the sideband asymmetry has solely been attributed to the mechanical zero-point fluctuations. The authors will find that this is not the case after a quick perusal of the relevant literature; alternative interpretations have been discussed at length.
- The authors assert that Eq (1) is simply asserted to be true, where in fact this expression appears to me to be just an expression proportional to the Fourier transform of the position correlation function of the undamped quantum harmonic oscillator, easily calculated. The full sideband spectrum of an optomechanical system has been calculated in other places, though this is not it.
- The authors undertake some calculations which appear reasonably standard, though I have not followed them through in detail. I expect they are mostly correct, though am suspicious of them in more subtle points where the authors are unclear.
- The crux of this work is their assertion that the calculations in Ref 6 are incorrect. Given this claim, they should directly compare their results with that work and point out the supposed flaw in Ref 6. As it stands, and without having reproduced either calculation in full myself, I find Ref 6 much more clear and convincing.
  • validity: poor
  • significance: poor
  • originality: poor
  • clarity: poor
  • formatting: acceptable
  • grammar: acceptable

Author:  João Machado  on 2020-01-09  [id 701]

(in reply to Report 2 on 2019-12-22)
Category:
answer to question
reply to objection

The referee presents unsubstantiated and rather vague criticism of our manuscript, which is largely based on arguments against statements not present in our manuscript and seemingly personal bias. We urge the referee to read the manuscript thoroughly and elaborate a report that reflects higher scientific standards. Particularly, the referee should pay attention to two major issues: the real goal of the manuscript, which is to show that backaction is responsible for sideband asymmetry; and careless attacks on the validity of the present work.

Regarding the latter, the referee states that "While I have not attempted to reproduce the authors’ calculations myself, I am skeptical as to their correctness.", " As it stands, and without having reproduced either calculation in full myself, I find Ref 6 much more clear and convincing." and "The authors undertake some calculations which appear reasonably standard, though I have not followed them through in detail. I expect they are mostly correct, though am suspicious of them in more subtle points where the authors are unclear.".

The referee repeatedly dismisses our findings, and yet is unable to point to any precise instance where the referee believes that we are wrong or unclear. We find that such statements lack any scientific basis and preclude all healthy scientific discussion. Further, the statements made by the referee that the referee has not attempted to reproduce nor follow our work in detail while judging it as unsound, constitutes a poor scientific practice. If the referee has spotted any mistake or flaw in our work, the referee should point what and where exactly the error is, for which we would be considerably grateful. The same holds for the points where the referee believes to be unclear.

Regarding the purpose of the article, this is repeatedly stated throughout the manuscript:

» abstract-'we find that the asymmetry arises from the backaction caused by the probe and the cooling lasers';

» introduction, end of page 2: 'we show that ZPM does not contribute to the asymmetry and that SA naturally arises from the backaction between the cavity and the mechanical oscillator';

» end of subsection 3.1, page 7: 'As backaction already occurs at the classical level, Eq.(23) shows a classical origin for SA.';

» end of subsection 3.2, page 8: '... backaction leads to an asymmetry in the spectrum.";

» conclusion, beginning of page 9: ' we have shown that SA arises from the backaction caused by the laser drive, and that no ZPM contributes to the asymmetry';

» conclusion, end of page 9: 'We stress once more that there are no quantum features in SA. For a linear system, the noise response function is identical in both classical and quantum descriptions, and as Stokes and anti-Stokes processes provide different amplitudes for the sidebands, an asymmetry naturally emerges.'

Nevertheless, the referee insists on contradictory views about our manuscript, none of which has any solid anchoring in our manuscript:

"The crux of this work is their assertion that the calculations in Ref 6 are incorrect." - The purpose of our manuscript was not to assert that Ref.6 was wrong. We believe that Ref.6 contains a mistake, and we mention it once in the body of the manuscript. Nevertheless, the emphasis given by the referee on Ref.6 does not reflect its importance to our present manuscript, nor does Ref. 6 stand out from the other references. We thus find that the insistence of the referee with Ref.6 is not due to any scientific reason.

"Sideband asymmetry ... has been attributed to zero-point fluctuations of the mechanical or electromagnetic modes, or due to correlation between backaction noise and the imprecision noise of the detector.
The present paper appears to support the latter interpretation" - There is no statement present in the paper that supports such interpretation.

Reply to the referee's remaining concerns:

"The spectral densities are defined in an unclear manner, presenting a spurious distinction between the time and frequency domain, where in fact they can be related in a well-defined manner if one is sufficiently careful. This has been done in a number of other places." - There are 2 issues to be addressed here:

»1st: we do not understand what precisely is unclear to the referee. None of the spectral densities present in the article are new, and they have been defined before elsewhere. Further, we always work in the frequency domain, so we do not understand what exactly the referee means with " spurious distinction between the time and frequency domain". We ask the referee to point to specific equations or sentences which the referee believes to be unclear, so that we can provide a proper answer.

»2nd: the referee seems to suggest that defining spectral densities in a quantum context is a matter of meticulosity. This is simply not true, and the issues associated with defining quantum spectral densities are already known from the literature (for a good review on the subject, see J.D. Cresser, Phys. Rep. 94, 47 (1983))

Related to this later issue, the referee states:
"The authors assert that Eq (1) is simply asserted to be true, where in fact this expression appears to me to be just an expression proportional to the Fourier transform of the position correlation function of the undamped quantum harmonic oscillator, easily calculated."

Our reply: The referee seems to have overlooked an entire section (Section 2) discussing the issues with the aforementioned spectral density (pages 3 & 4 of our manuscript). Further, the "easily calculated Fourier transform of the position correlation function of the undamped quantum harmonic oscillator" is not as well-defined as the referee seems to believe (once again, see J.D. Cresser, Phys. Rep. 94, 47 (1983)).

And at last, the referee states:
"The authors claim ... that the sideband asymmetry has solely been attributed to the mechanical zero-point fluctuations."

Our reply: We do not make such claim. Once the referee reads our manuscript carefully, the referee will find that we discuss other interpretations of sideband asymmetry as well:

»'Alternative interpretations and explanations such as interference between different noise channels [6,15] and laser phase noise [16] have been discovered': page 2 of our manuscript

»'The interpretation of SA differs for different operator orders (arising from different detector models' - page 2 of our manuscript

Further, the referee's comments about Ref.6 answered above contradict the referee's own statement.

---

## Round 5 · Referee Report · Anonymous · 2020-3-11

Report
Upon resubmission, the authors have made a number of changes to the main text. This includes shifting large sections of text to the supplementary information, and including a brief comment on experiments that leverage single photon detection. I think these changes improve the readability of the paper.
In the previous submission, referee 2 raised a number of criticisms of this work. I can certainly understand his/her perspective on certain aspects, in particular, that the subtle interpretation of sideband asymmetry has been the subject of numerous detailed papers over the last decade. For example, there is the PRA by by Kjetil Børkje (Ref 18) which details the interpretation of sideband asymmetry. There is also the review article by Clerk et. al. (Ref 14) which provides an extremely detailed account of quantum measurement theory. However, without going through the detailed calculations, I can't find any specific errors/contradictions in this manuscript that would negate its claims.
Quite generally, I think this subject is interesting and timely. As a result, I think it should be published in SciPost.
Author: João Machado on 2021-03-22 [id 1324]
(in reply to Report 2 on 2020-03-11)We thank the referee for the comments regarding our previous statement on the dependence of the mechanical quality factor with temperature. The linear change in the mechanical dissipation with temperature was meant for the limit of very small cooperativities C -> 0, and it does not hold for the general case. We have corrected this statement and made a more careful derivation of how the mechanical quality factor should behave with temperature in order to describe the experimental observations. We cannot guarantee that this dependence fully accounts for the observed asymmetry in all experimental situations, but it does play a role as the mechanical damping rate can vary by 3 orders of magnitude over the relevant temperature range. Alongside the discussion in our manuscript, please see also the reply to report 3 regarding the comparison with the experimental observations. We kindly ask the referee to put forward any further questions or concerns the referee might have concerning any issue of our manuscript.
Strengths
In helping to make the clear the relationship between classical and quantum linear systems, the manuscript makes a valuable contribution to a topic of current interest.
Weaknesses
None
Report
I think the authors have done a good job of improving the manuscript based on the suggestions of the previous referees. I recommend the manuscript for publication.
Requested changes
none
Author: João Machado on 2021-03-22 [id 1326]
(in reply to Report 1 on 2020-02-14)
We fully understand the comment from the referee, but
First, reference 6 does not treat the issue of the classical nature of the measured spectrum for linear systems, but merely analyses the symmetric spectral density for a multitone driven system. We show in section 4 of our manuscript that when fully including the effects of the linear interaction, the multimode and multitone results differ due to interference between the multiple tones, which enables to differentiate the source of the asymmetry. This result has not been reported before, and we believe it is valuable on its own.
Second, we have absolutely no intention of dismissing previous work, solely to carefully analyse their claims. We have now stated this up front in the introduction. We have also rederived and presented detailed calculations of previous works, and discussed the results in section 4, as well as highlighted the differences and consequences for the interpretation of sideband asymmetry. We hope to bring full clarity to the subject and we kindly ask the referee to speak up if this is not the case.
Third, we cannot possibly guarantee that the temperature dependence of the mechanical damping rate fully explains the temperature dependence of the asymmetry for every experimental case, but it does play a role as its value can change by 3 orders of magnitude. We now discuss more carefully in our manuscript how the the mechanical damping rate must depend on temperature to describe the experimental observations. As the referee must be aware, we have limited information about every experimental measurement of sideband asymmetry, but we have done to compare our results with the experiments. Along with the related discussion in our manuscript, please see our reply to report 3.
Anonymous on 2020-04-20 [id 801]
Having read the latest reports from the other referees, I realize that I had not looked carefully enough at the manuscript in my first report. In particular, I was not aware that the issue of the classical nature of the measured spectrum for linear systems has already been treated in detail (especially reference 6).
For what the manuscript claims, the presentation is inadequate. As referee 2 states, the primary topic of the manuscript has already been treated in detail. Thus the novelty of the manuscript relies on the authors claim that multiple previous papers have significant errors, including calculations errors. This is presumably the reason that, in the introduction, they are so dismissive of previous work on the subject. The manuscript also contains another unusual claim, which is that the previous experimental observations of the expected temperature-dependent asymmetry is not due to the temperature dependence found in previous derivations, but due only to the temperature dependence of the mechanical damping rate. Both claims are sufficiently improbable that they require a detailed justification. This is why the presentation is wholly inadequate. As a previous referee states, if the authors believe that there are errors in previous papers sufficiently significant as to render the conclusions incorrect, one option is to write comment specifically on one of those papers, redoing the calculations correctly so that everyone can see exactly where the errors are. If the authors wish to write a single paper in which the claim significant errors in a number of papers they need to provide supplementary material, or appendices that gives the details of the calculations to make clear exactly what the errors are. Also, the claim about errors in previous works should be up front in the introduction, since it is a major part of the work; it is the claimed reason that previous work has come to a very different conclusion about the form of the output spectra.
The output spectrum obtained by the authors leads to the conclusion that the temperature dependence of the asymmetry has a completely different origin than previously thought. This also requires a much more detailed comparison with previous work, this time experimental. The claims of previous errors, and the claims of a new origin for the temperature dependence, as far as I can tell, go together. If one is incorrect so is the other, most likely. That’s why the authors should be checking their claims against previous work in much more detail.
So, as with the previous referees, I do not recommend publication without significant additions to the manuscript.
Matthew Davis on 2020-04-21 [id 802]
(in reply to Anonymous Comment on 2020-04-20 [id 801])As the editor, I am verifying that the comment above is by the author of report 1 from the second round of refereeing.
Matthew Davis on 2020-04-16 [id 794]
From the editor This is a further comment delivered by email by the author of report 1 from the first round, and report 2 of the second round of refereeing. It is cut and paste as a comment by the editor with the author's permission. I have clarified that they are discussing the comments in report 2 from the first round, and report 3 from the second round of refereeing.
From the referee

---

## Round 5 · Referee Report · Anonymous · 2020-4-5

Report
As previously stated, there has been prior work on the necessity of considering the symmetrized noise spectral density in order to faithfully describe measurement outcomes (e.g., Ref 6). It is even clearly stated in the standard RMP in the field (see Ref 14). There have also been calculations of optical spectra based upon the normally-ordered first-order correlation function (e.g., Ref 11), as measured via photodetection. While it is accepted that this distinction is not always clearly made, it clearly has been previously discussed at length, particularly in the context of sideband asymmetry.
Therefore, the text around Eq (1) appears to still be largely incorrect.
Also as previously stated, Eq. (1) is still just the Fourier transform of a position correlation function; whether this quantity is interpreted as a measured spectral density is a separate matter. Further, Eq (1) confusingly conflates the calculation of spectra of scattered optical fields with the calculation of a noise spectrum of the mechanical oscillator itself.
I accept Eq (2) though it is misleading to represent this as a novel observation. It would also be helpful to make it clear that (X(\omega))^\dagger and X(-\omega) are the same. This quantity is essentially the Fourier transform of the symmetrized quadrature correlation function; this is the connection between the time and frequency domains referred to in the earlier report.
The present work here still seems to me to be very closely related to Ref 6, in particular. Both papers purport to calculate the symmetrized spectral density of an optical field quadrature coupled to a thermal oscillator.
I do not believe it is reasonable to expect a referee to reproduce lengthy calculations in detail; one can only judge from the clarity of the presentation and the plausibility of the conclusions. I find the present paper lacking in both respects. The issues with the introductory material have already been addressed.
I find the conclusion that observed sideband asymmetry being merely a coincidence of the simultaneous modification of the intrinsic damping with temperature to be unlikely. In order to make this claim more reasonable, experimental data should be used to specify how much this damping would have to vary with temperature, and assess if this is consistent with observed experimental variations in temperature-dependent damping.
Finally, given that Appendix C seems to claim that multiple published papers are in error, perhaps it would be more appropriate to write comments on these papers in order to elicit a response from the original authors of these papers. It seems more reasonable than expecting a referee to attempt to reproduce calculations across multiple apparently conflicting papers.
Author: João Machado on 2021-03-22 [id 1323]
(in reply to Report 3 on 2020-04-05)
We understand the scepticism of the referee, but we must disagree with the referee.
We ask the referee to keep an open mind and not to base the final judgement on the referee's own believes. Statements such as "one can only judge from the clarity of the presentation and the plausibility of the conclusions" do not contribute to healthy scientific discussion and are simply an euphemism for judging the present work superficially and evaluating it based on whether our conclusions meet the referee's expectations or not. The referee is surely aware that this is not the only way to judge a manuscript and that "plausibility" does not equate to scientific validity. Further, there is nothing implausible about our conclusions: that the coupling between light and mechanics breaks the symmetry of the spectral sidebands' height.
Further, the referee says "I do not believe it is reasonable to expect a referee to reproduce lengthy calculations in detail". We have not asked the referee to reproduce any equation presented, merely to point to where the referee believes there is a mistake and to provide a logical explanation of why the referee believes our work to be incorrect. This has not been done, and questioning the validity of our work without any solid basis is not a scientific practice.
The major concern of the referee appears to be the comparison with the experimental observations of sideband asymmetry. There is nothing in the experimental observations of sideband asymmetry that contradicts our conclusions, and a discussion on this can be found below. We have also expanded on the comparison with the experiment in our manuscript.
The referee states "I find the conclusion that observed sideband asymmetry being merely a coincidence of the simultaneous modification of the intrinsic damping with temperature to be unlikely."
» First, our conclusion is not that sideband asymmetry is due to some modification of damping with temperature. What we conclude is that sideband asymmetry is due to the optomechanical damping created by the interaction. This is repeatedly stated throughout the manuscript.
» Second, we do not see any "coincidence" of any kind in our results, and we cannot possibly guess why the referee believes our conclusion to be unlikely without additional explanations. As stated before, there is nothing implausible about our conclusions: that the coupling between light and mechanical motion breaks the symmetry of the spectral sidebands.
» Third, the temperature dependence of sideband asymmetry to be caused by the variation of
the intrinsic damping with temperature is far from unlikely as the intrinsic damping can vary by 3 orders of magnitude for the typical experimental temperature ranges. This is now discussed in the manuscript.
The referee says "In order to make this claim more reasonable, experimental data should be used to specify how much this damping would have to vary with temperature, and assess if this is consistent with observed experimental variations in temperature-dependent damping.". We agree that an experimental comparison would strengthen our conclusion, but as the referee must be aware, in most cases, direct comparison with the experimental observations is not possible, because the measured asymmetry is not reported as a function of
temperature, and the temperature dependence of the physical parameters is not determined. To be able to present a flawless comparison of the temperature dependence for every single experiment, we would need to have access to the necessary experimental data (which we do not have), but it is our hope that our research motivates a dedicated study of this. Nevertheless, there is some experimental data indicating that the change in asymmetry due to temperature dependent damping is a reasonable feature. For example, ref.5 reports an
increase by 3 orders of magnitude in the intrinsic mechanical damping with the increase in cavity temperature, which for small cooperativities results in an increase by the same magnitude in the asymmetry as the temperature decreases. This is a rather dramatic effect that should not be overlooked. Ref.5 reports g_0=910kHz and k=488kHz. For a single weak readout laser (coherent state with ~0.1 photons), the asymmetry at 18K (gamma~1kHz) predicted by Eq.(11) of our manuscript would be ~4.25, while at 210K (gamma~1MHz),
the asymmetry would drop to 0.0014, which are realistic values. Note however that the asymmetry reported in ref.5 was not measured with homodyne/heterodyne technique discussed in the manuscript.
Although we do not have access to the real experimental data, we can make some simple predictions for the experiment reported in ref.6. Ref.6 reports the sideband ratio as a function of sideband height. The largest sideband ratio reported was of ~4.81 (value obtained by visual inspection of the presented dataplot) with the corresponding lowest value of ~1.38 (also obtained by visual inspection) for the same injected noise level. According to our estimations, the cooperativities would have to be ~0.66 at 20mK and ~0.16 at 200mK,
which corresponds to an increase by a factor of ~4.1 for the intrinsic mechanical damping (assuming no other parameter changes with temperature). The variation of intrinsic mechanical damping with temperature is not reported in the paper, but an increase by a factor of ~4.1 over this temperature range is nothing extraordinary. Similar changes in mechanical damping rate have already been reported in the literature (see for example, Appl. Phys. Lett. 107, 263501 (2015), where a mechanical Q of 1.27e8 at 14mK is reported with the same membrane having a Q<4e7 at 200mK; this is an increase >3x over approximately the same temperature range). Note once more that we do not have access to the experimental data, and so the accuracy of these predictions is limited.
An overview of other experimental observations of sideband asymmetry and the associated hindrances in the comparison follows:
Ref.4: The asymmetry is not directly presented, but a modified version of the asymmetry. This modified version is not represented as a function of temperature but as a function of the estimated phonon number based on the asymmetry itself. We have no way of determining the real temperature profile associated with the data. Further, the intrinsic mechanical damping variation with temperature is not reported.
Ref.7: The measured raw spectra are presented, but not the asymmetry values associated with each spectrum. To be able to obtain these, we would need to integrate the data over the frequency range. Further, the spectra are presented as a function of cooperativity, not temperature. Again, the intrinsic mechanical damping variation with temperature is not reported.
Ref.8: The sideband ratio is measured for different temperatures, and the estimated temperature by the asymmetry is plotted against the cryostat temperature. There are at most 4 points in a loglogplot, where not all points reach the fitted curve. It is thus not possible for us to determine the precise values with the naked eye. Further, the cooperativities of the modes are not reported, and there is not enough information to determine them, and so we cannot estimate the effects of backaction. Again, the intrinsic mechanical damping variation with temperature is not reported.
Ref.9: The asymmetry is reported as a function of the cooling laser power. As before, the cooperativities are not reported, and there is not enough information to determine them, and so we cannot estimate the effects of backaction. Again, the intrinsic mechanical damping variation with temperature is not reported.
Replies to the remaining referee's comments:
The referee states "there has been prior work on the necessity of considering the symmetrized noise spectral density in order to faithfully describe measurement outcomes (e.g., Ref 6)". As far as we are aware, our explanation for the necessity of symmetrised spectral functions is not found anywhere else. Further, a careful reading of ref.6 shows that it does not provide an explanation for the necessity of symmetrised spectral densities. This article is about the role of interference of the noise channels in the asymmetry and the experimental observation of sideband asymmetry. Here the term backaction is a misnomer.
The referee states that "the text around Eq (1) appears to still be largely incorrect". It is not. Discussions about the role of zero-point motion on the asymmetry can be easily found in the literature (c.f. refs. 4 or 8). Further, we have also discussed the relevant literature dealing with symmetrised spectral densities mentioned by the referee in the introduction (see "Alternative interpretations and explanations such as interference between different noise channels [6, 15]..."). The referee seems once again to have overlooked this.
The referee says "Also as previously stated, Eq. (1) is still just the Fourier transform of a position correlation function". The fact that Eq.(1) is the Fourier transform of a position correlation function was never questioned, only its physical meaning. The fact that one is able to compute the Fourier transform of a correlation function is not intertwined with whether the result of such computation represents a physically measured quantity or not. As discussed in the manuscript, there are several possible Fourier transforms of correlation functions, but not all of them represent a physical measurement. Indeed "whether this quantity is interpreted as a measured spectral density is a separate matter", but the whole significance of sideband asymmetry lies precisely on this matter.
The referee states "Eq (1) confusingly conflates the calculation of spectra of scattered optical fields with the calculation of a noise spectrum of the mechanical oscillator itself.". It does not. Eq.(1) and the discussion surrounding it merely reflects statements which can be found in the literature (see for example the introduction of ref. 8).
The referee states "I accept Eq (2) though it is misleading to represent this as a novel observation". We have not claimed anywhere in the manuscript that symmetrised spectral densities are a novelty. Eq.(2) is merely a starting point for spectral calculations, and the discussion surrounding it only aims at arguing why a symmetrised spectral density should be preferred when computing the spectrum for quadrature measurements. Once again, our main point is the nature of the asymmetry.
"It would also be helpful to make it clear that (X(\omega))^\dagger and X(-\omega) are the same.". We think that the standard property of taking the complex conjugate of the Fourier transform (i\omega -> i (-\omega)) is well-known and it does not require an explanation of its own.
Appendix C serves merely as a reference of sensitive points in the literature, and to help place this manuscript in context. We obviously did not expect (nor asked) the referee to reproduce any calculation from the appendix, and the appendix should not divert the focus from the main message of the manuscript, which is that the optomechanical damping leads to sideband asymmetry.

---

## Round 5 · List of Changes

List of changes:
» The Fourier coefficients of the optical response function in Eq.8 and the optical response for the case where each sideband is driven separately are now displayed in Appendix A.
» The multi-tone case was moved to a separate appendix (App.B)
» Briefly discussed the photon counting detection experiments
- R. Riedinger, S. Hong, R. Norte, J. A. Slater, J. Shang, A. G. Krause, V. Anant, M. Aspelmeyer, and S. Gr\"{o}blacher, {\it Non-classical correlations between single photons and phonons from a mechanical oscillator}, Nature {\bf 530}, 313 (2016), \doi{10.1038/nature16536}
- J. Cohen, S. Meenehan, G. MacCabe, , S. Gr\"{o}blacher, A. H. Safavi-Naeini, F. Marsili, M. D. Shaw, and O. Painter, {\it Phonon counting and intensity interferometry of a nanomechanical resonator}, Nature {\bf 520}, 522 (2015), \doi{10.1038/nature14349}
in section 4 (conclusion). It now reads:
Further, our analysis is focused only on the power spectral density obtained with linear detections methods. There are other methods closely related to SA which also make use of the term ``sideband asymmetry'', such as the difference between the photon count rates when an optomechanical system is driven with short pulses at the red and blue sidebands [CITATION], that are outside the domain of the present work. Because single photon detection is a method that does not exhibit the operator order issues raised in Sec.2, and because only the instantaneous count rate is measured instead of the frequency spectrum (for weak driving and short enough pulses, the effects of backaction and noise are negligible), our conclusions do not apply to these single photon detection methods. The quantum description of the photon count detection scheme does indeed display a $n/(n+1)$ imbalance and can be used to directly determine the phonon occupancy. Nevertheless, it would be interesting to understand how the two methods are related to each other.
» The new structure of the article was updated in the introduction:
In this article, we start by discussing the problems with the standard interpretation of SA, their connection to how measurements are performed, and the role of ZPM in the spectrum (Sec. 2). We proceed to compute the response function for a system composed by two driven optical modes and a mechanical one (Sec. 3). Considering the symmetric noise power spectral density for both cases, we show that ZPM does not contribute to the asymmetry and that SA naturally arises from the backaction between the cavity and the mechanical oscillator. Additional details of the calculations can be found in App. A. For comparison with other measurement methods, we also analyse the case of a system composed by two modes (cavity + mechanical), but driven with multiple tones in App. B. Finally, we present a concise review of previous analyses of SA in App. C.

---

## Editorial Decision

unknown